# Variable PD-1 glycosylation modulates the activity of immune checkpoint inhibitors

Chih-Wei Chu[1], Tomislav Čaval[1], Frederico Alisson-Silva[1], Akshaya Tankasala[1], Christina Guerrier[1], Gregg Czerwieniec[1], Heinz Läubli[2], Flavio Schwarz[1]

**Monoclonal antibodies targeting the immune checkpoint PD-1 have provided significant clinical benefit across a number of solid tumors, with differences in efficacy and toxicity profiles possibly related to their intrinsic molecular properties. Here, we report that camrelizumab and cemiplimab engage PD-1 through interactions with its fucosylated glycan. Using a combination of protein and cell glycoengineering, we demonstrate that the two antibodies bind preferentially to PD-1 with core fucose at the asparagine N58 residue. We then provide evidence that the concentration of fucosylated PD-1 in the blood of non–small-cell lung cancer patients varies across different stages of disease. This study illustrates how glycoprofiling of surface receptors and related circulating forms can inform the development of differentiated antibodies that discriminate glycosylation variants and achieve enhanced selectivity, and paves the way toward the implementation of personalized therapeutic approaches.**

## Introduction

PD-1 is a type-I transmembrane glycoprotein expressed in immune cells, predominantly in T cells, that functions as a checkpoint: engagement of PD-1 by its ligands PD-L1 and PD-L2 suppresses TCR-driven cell activation and induces apoptosis (1). This mechanism can be detrimental in the tumor microenvironment, where a combination of exhausted T cells displaying PD-1 and PD-L1 expression by cancer and myeloid cells causes ineffective antitumor immune responses (2). Consequently, monoclonal antibodies that target PD-1 or its ligands and prevent their interaction have been developed as anticancer therapeutics (3, 4, 5, 6). Whereas immune checkpoint inhibitors have revolutionized cancer treatment across many tumor indications and demonstrated the usefulness of interjecting the PD-1:PD-L1 axis, only a fraction of patients benefit from these drugs and develop durable clinical responses (7, 8).

Many mechanisms may contribute to limit this approach: the lack of appropriate T cell priming, their exhaustion status, the presence of stromal factors that prevent penetration of T cells to the tumor parenchyma, and the relative abundance of lymphoid and myeloid cells with immunosuppressive properties (9, 10). Critical information about the PD-1 pathway and the mechanisms of signaling blockade may also still be unavailable. Moreover, the intrinsic properties of anti-PD-1 antibodies might be responsible for differences in efficacy or safety observed in the clinical settings.

PD-1 and PD-L1 are regulated both at the transcriptional and at the post-translational level. PD-1 is induced upon TCR-mediated activation and decreases with antigen clearance, but it is maintained on antigen-specific T cells in chronic settings such as cancer (11). PD-1 stability at the cell surface is controlled by ubiquitination that leads to protein degradation (12) and by glycosylation that may occur at four asparagine sites (N49, N58, N74, and N116). CRISPR-based screening identified Fut8, which encodes for a fucosyltransferase that adds $\alpha$1,6 core fucose to N-glycans, as a mechanism that regulates the cell surface expression of PD-1 by modification of glycans at sites N49 and N74 (13). In line with this observation, T cells exposed to a fucose inhibitor produce stronger antitumor responses (13, 14). Similarly, PD-L1 expression and stability are regulated at the post-translational level by glycosylation and ubiquitination (15). PD-L1 glycosylation is also critical for interaction with PD-1; in contrast, PD-1 glycosylation appears to be dispensable for binding to PD-L1 as its glycosylation sites are distant from the ligand binding interface (12). Interestingly, some antibodies, including cemiplimab and camrelizumab, interact with a region surrounding the N58 glycosylation site and require the N58 glycan for efficient binding (16, 17). This property is not shared by other antibodies, such as nivolumab and pembrolizumab, that have partly overlapping binding epitopes interacting with the flexible N- and C'D-loops of the PD-1 molecule, respectively (18, 19, 20, 21).

In addition to the transmembrane forms, soluble PD-1 and PD-L1 variants (sPD-1 and sPD-L1) can be generated by protease-based cleavage and accumulate in blood (22, 23). Although the function of the released protein variants is not fully established, it has been reported that sPD-L1 retains inhibitory capacity (24) and elevated

[1]InterVenn Biosciences, South San Francisco, CA, USA   [2]University of Basel, Department of Biomedicine, and University Hospital Basel, Division of Oncology, Basel, Switzerland

Correspondence: flavio.schwarz@venn.bio

sPD-L1 levels have been associated with advanced disease and worse prognosis (25). sPD-1 appears to be less informative as a prognostic or predictive biomarker (22). Notably, the glycosylation status of sPD-1 or sPD-L1 has not been determined.

Here, we observed that camrelizumab and cemiplimab specifically interact with a fucose moiety within the N58 glycan of PD-1. As fucosylation is increased in cancer and fucosylated biomarkers in blood have been associated with a lack of benefit of immune checkpoint inhibitor therapy (26, 27), we investigated the influence of fucosylation on PD-1 in antibody activity by a combination of protein- and cell-based assays. Furthermore, we characterized fucosylation of sPD-1 in the serum of individuals with non–small-cell lung cancer (NSCLC). These data illustrate a path toward the development of personalized immunotherapeutic approaches for cancer treatment and highlight the potential of glycosylation analysis of targets to guide the development of differentiated therapeutics.

## Results

### Glycosylation-dependent anti-PD-1 antibodies interact with the core fucose

A number of monoclonal antibodies have been approved as anticancer treatment in a broad range of indications (Table S1) with clinical efficacy and toxicity profiles that may be related to the intrinsic properties of the molecule. A subgroup of these antibodies, including camrelizumab and cemiplimab, have been recently identified for their dependency on the N58 glycosylation for binding (16, 17). To elucidate the detailed binding characteristics of camrelizumab and cemiplimab to PD-1 and the PD-L1 mechanism,

we inspected the available structures of antibody:PD-1 complexes. In line with what was previously described, we observed that camrelizumab and cemiplimab bound to a similar region of PD-1 in the proximity to the N58 glycosylation site, whereas pembrolizumab and nivolumab recognized different epitopes (Fig S1A and B). In particular, the presence of the N58 glycan of PD-1 in the camrelizumab:PD-1 complex allowed us to visualize the interactions between the VH and the glycan (Fig 1A and B). In addition to the previously described interactions between the HCDR1 and HCDR2 and the core glycan, we observed that residues S30 and S31 of the CRD1 and S52, G53, and G54 of the CRD2 engaged the α1,6 core fucose (Fig 1B). Although the PD-1 N58 glycan was not visible in the complex with cemiplimab, the antibody appeared to bind to PD-1 in a similar orientation as camrelizumab, suggesting the N58 glycan composition may influence the binding and activity of both antibodies (Fig 1C).

### Core fucose of the PD-1 N58 glycan is a key determinant for camrelizumab and cemiplimab binding

To directly test whether the core fucose of the PD-1 N58 glycan was involved in antibody binding, we used both protein- and cell-based assays. First, we generated three recombinant PD-1 variants by expressing the extracellular region of PD-1 in Expi293 cells in the presence or absence of the fucose inhibitor 2-fluoro-peracetyl-fucose (2FPF), and by producing a N58Q mutant protein (Fig 2A). Mass spectrometry analysis confirmed that the glycoforms at the N58 site of the PD-1 and PD-1 NF (expressed in the presence of the fucose inhibitor) variants were comparable and differed only for the presence or absence of fucose (Figs 2B and S2, S3, and S4). Binding of camrelizumab and cemiplimab to the three PD-1 variants was examined by a biolayer interferometry assay, along with nivolumab and pembrolizumab as controls (Figs 2C and S5). We

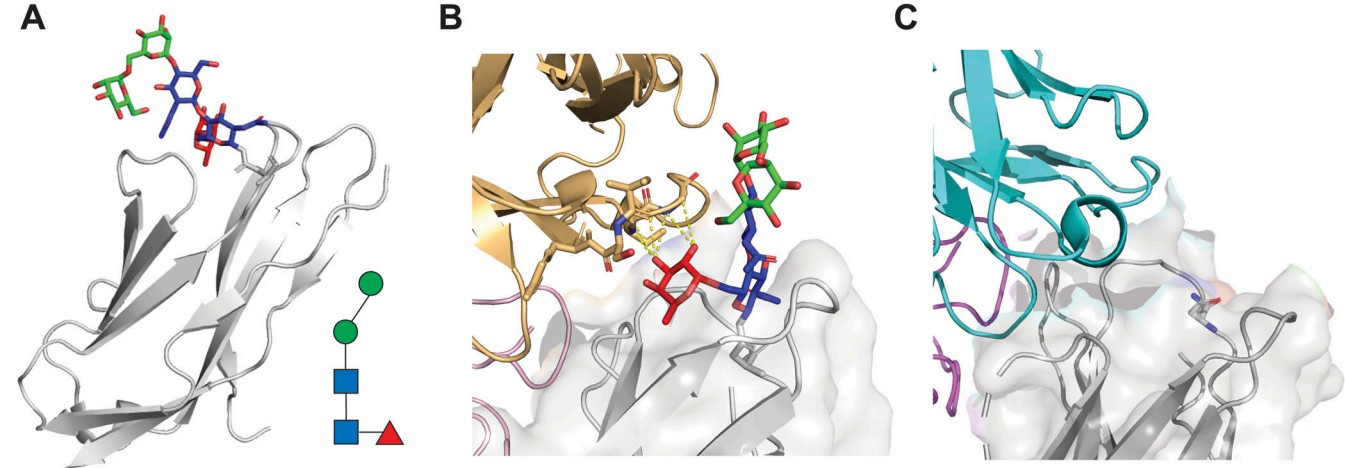

**Figure 1. Binding interface between PD-1 and the anti-PD-1 antibodies camrelizumab and cemiplimab includes the core fucose of the PD-1 N58 glycan.**
**(A)** Structure of PD-1 (gray) with the glycan linked to the asparagine N58 of the BC-loop of PD-1. The glycan visible in the crystal structure (PDB 7CU5) includes a fucose unit, two *N*-acetylglucosamine units, and two mannose units. The glycan structure is also represented with geometric symbols as convention (red triangle: fucose; blue square: *N*-acetylglucosamine; and green circle: mannose). **(B)** Structure of PD-1 in complex with camrelizumab (PDB 7CU5) indicating potential interactions between the core fucose of PD-1 N58 glycan and the heavy chain of the antibody (residues S30 and S31 of CRD1 and S52, G53, and G54 of CRD2). The VH chain is indicated in gold, and the VL chain is in pink. **(C)** Structure of cemiplimab in complex with PD-1 (PDB 7WVM). The N58 residue of PD-1 is represented as sticks. Cemiplimab binds to PD-1 in a similar orientation as camrelizumab, suggesting the N58 glycan structure may influence affinity. VH is in cyan, and VL is in purple.

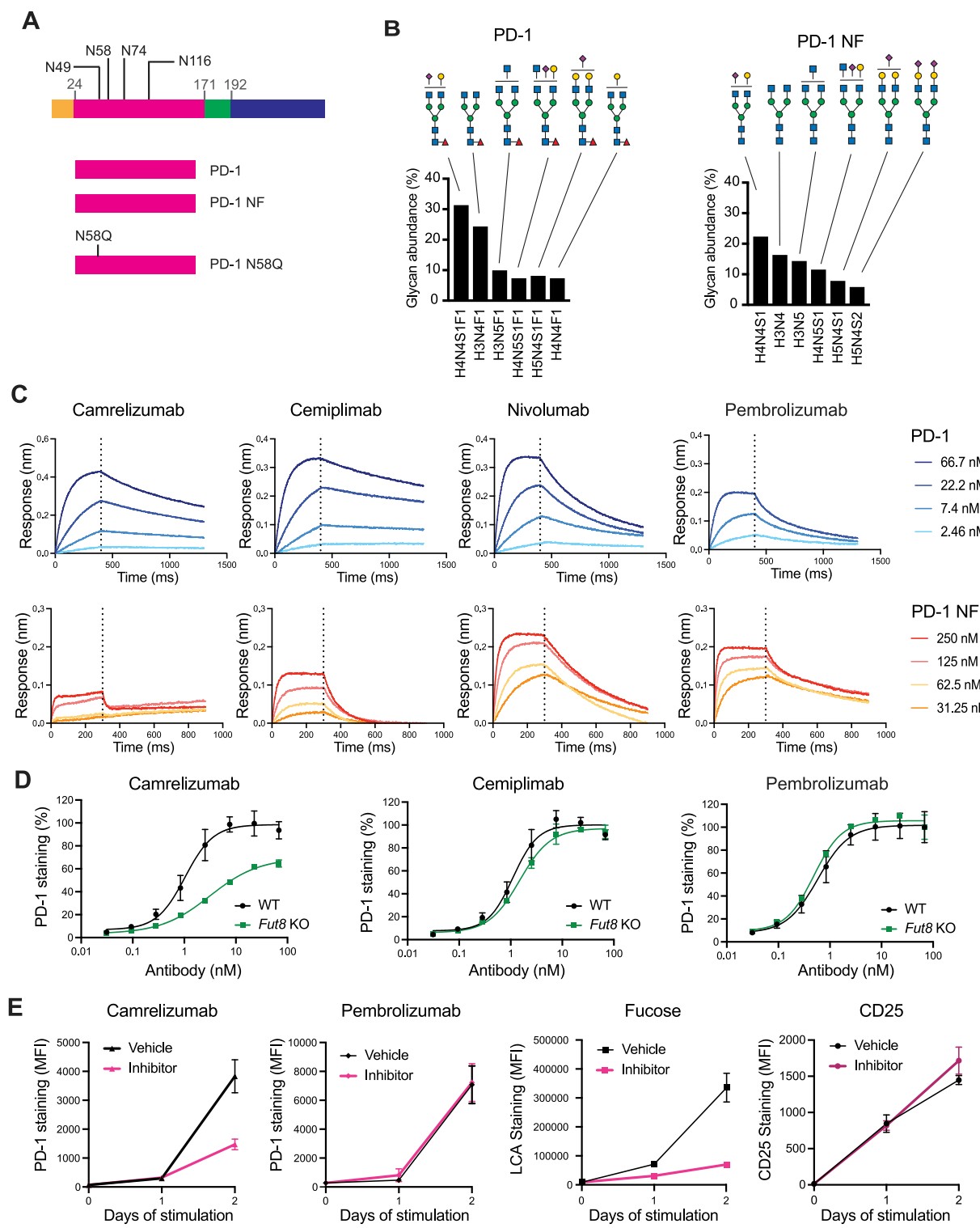

Figure 2.  **Binding of camrelizumab and cemiplimab to PD-1 depends on the core fucose of the N58 glycan of PD-1.**
**(A)** Schematic diagram of the PD-1 proteins indicating the signal peptide (in orange), extracellular domain (in pink), transmembrane domain (in green), and intracellular region (in blue), as well as the four N-linked glycosylation sites. **(B)** Relative abundance of the glycoforms found at the N58 site of recombinant PD-1 and non-fucosylated PD-1 (PD1 NF). Glycan structures are comparable and differ only by the fucose moiety (red triangle). Glycan composition is indicated at the bottom of the graph (H, hexose; N, N-acetylhexosamine; S, sialic acid; and F, fucose) and represented with symbols at the top. **(C)** Binding profiles of anti-PD-1 antibodies to recombinant PD-1 proteins. The dotted line marks the separation between the association and the dissociation cycle. **(D)** Binding of anti-PD-1 antibodies to CHO-K1 (WT) or CHO-K1 *Fut8* knockout (*Fut8* KO) cells expressing chimeric receptors including the extracellular and transmembrane domains of PD-1 and intracellular GFP. Lack of FUT8 expression did not alter the binding profile (EC50 value) of pembrolizumab (*P* = 0.25) but resulted in reduced binding of camrelizumab (*P* = 0.02). Binding of cemiplimab was only marginally

observed a >100-fold difference in the affinity of camrelizumab to PD-1 and PD-1 N58Q, in line with previous reports (Table 1). Strikingly, the difference in KD between PD-1 and the N58Q mutant was completely recapitulated by PD-1 NF (>100-fold change in the KD value), indicating that the contribution of the N58 glycan to the antibody binding was largely provided by the core fucose. Similarly, binding of cemiplimab to PD-1 was compromised by the absence of fucose, although not to the same extent as camrelizumab (29-fold change in $K_D$). As expected, binding of pembrolizumab or nivolumab to the PD-1 variants was not substantially affected by changes in the glycosylation structure.

Next, to further test the hypothesis that core fucose was a key determinant for binding, we analyzed antibody binding to cell surface PD-1 expressed in wild-type or Fut8 KO CHO-K1 cells. The absence of core fucose reduced binding of camrelizumab (three-fold change in the EC50 value) and cemiplimab (1.3-fold change in the EC50 value) but did not alter binding of pembrolizumab (Fig 2D).

Lastly, we assessed PD-1 detection by camrelizumab in primary human CD8 T cells activated in the presence or absence of a fucose inhibitor. Naive CD8 T cells expressed no or relatively low levels of PD-1 and CD25 (Fig 2E). As expected, CD3/CD28 stimulation induced CD25 expression within 24 h and PD-1 expression within 48 h. Activation in the presence of the fucose inhibitor resulted in a substantial decrease in PD-1 detection by camrelizumab but did not affect recognition by pembrolizumab. CD25 expression was not altered by the presence of the fucose inhibitor during cell activation. Altogether, these data indicate that both fucosylated and non-fucosylated PD-1 can be expressed on the surface of T cells and that binding of camrelizumab and, to a lesser extent, cemiplimab depends on core fucosylation of the N58 glycan of PD-1.

### PD-1 fucosylation affects PD-L1 blocking efficacy of camrelizumab and cemiplimab

Next, we sought to understand whether the glycan structure of PD-1 had an impact on the ability of anti-PD-1 antibodies to block interactions with PD-L1. First, we assessed whether the fucosylation status of PD-1 modulated the interaction with its ligand and observed no differences in PD-L1 binding to PD-1 in CHO-K1 cells regardless of *Fut8* expression (EC$_{50}$ values were 1.0 nM for CHO-K1 cells and 1.7 nM for CHO-K1 *Fut8* KO cells) (Fig S6). However, whereas ligand blocking profiles were overlapping in CHO-K1 cells, camrelizumab and cemiplimab were less effective in preventing PD-L1 interactions in *Fut8*-deficient cells (calculated IC50 values were 5.6 nM for camrelizumab, 2.6 nM for cemiplimab, and 0.9 nM for pembrolizumab) (Fig 3A).

Next, to further investigate the impact of fucosylation on the blocking efficacy of anti-PD-1 antibodies, we monitored their capacity to impair binding of PD-L1 to plate-coated PD-1 variants (Fig 3B). Inhibition of PD-L1 binding to PD-1 was similar for cemiplimab, camrelizumab, and pembrolizumab. However, blockade of PD-L1 binding was sensitive to both fucosylation and glycosylation at the N58 site, with calculated IC50 values 1.5 nM for camrelizumab, 0.6 nM for cemiplimab, and 0.5 nM for pembrolizumab for PD-1 NF, and 18 nM for camrelizumab, 1.4 nM for cemiplimab, and 0.7 nM for pembrolizumab for PD-1 N58Q.

Lastly, to determine whether the observed differences in ligand blockade had functional consequences in modulating PD-1 signaling, we employed a reporter assay that uses suppression of luciferase expression as a readout of PD-1:PD-L1 interaction. Consistently with our previous observations, we did not detect differences in the blocking ability of the anti-PD-1 antibodies in untreated cells (Fig 3C). However, pretreatment of Jurkat-hPD-1 cells with the fucose inhibitor (Fig S7) revealed differences in the potency of the antibodies, resulting in a 2.8-, 2.4-, and 0.8-fold change of IC50 values for camrelizumab, cemiplimab, and pembrolizumab, respectively. Treatment with the fucosylation inhibitor did not alter PD-1 expression on the cell surface (Fig S8). The lower overall signals detected in Jurkat cells exposed to the fucose inhibitors might be related to the impact of fucose on other components of the immune synapse. Taken together, these data indicate that the blocking efficacy of the anti-PD-1 antibodies camrelizumab and cemiplimab is influenced by PD-1 fucosylation at the N58 site.

### Fucosylated PD-1 increases in the serum of late-stage lung cancer patients

As camrelizumab and cemiplimab are selective for fucosylated PD-1, we investigated whether there was variability in the fucose content of soluble PD-1 in the blood of cancer patients. To detect fucosylated PD-1, we developed a plate-based assay that employs plate-bound camrelizumab or pembrolizumab for capturing PD-1, followed by detection with polyclonal anti-PD-1 antibodies. The two assays were able to discriminate between fucosylated and non-fucosylated PD-1 (Fig S9). Surprisingly, similar binding patterns were observed with commercial kits. Using combinations of these assays, we were able to measure fucosylated PD-1 in the serum of a cohort of NSCLC patients (Table S2). Whereas there were no differences in the concentration of total serum PD-1 in different disease stages (Fig 4A), the fraction of fucosylated PD-1 appeared to increase in late-stage lung cancer (Fig 4B).

To investigate whether the glycosylation of serum sPD-1 impacts binding of anti-PD-1 antibodies to PD-1 on T cells, we determined camrelizumab binding to Jurkat cells expressing PD-1 in the presence of PD-1 or PD-1 NF, as surrogates of serum sPD-1. In line with what we had observed earlier, fucosylated PD-1 prevented

affected (*P* = 0.17). Antibody binding was calculated as a percentage of PD-1 expression in GFP-positive cells. Average values and SD of three independent experiments are represented. Interpolation was calculated using a four-parameter logistic function. **(E)** Binding of anti-PD-1 antibodies to human CD8 T cells. Cells were activated with CD3/CD28 beads in the presence or absence of a fucose inhibitor and tested for the expression of PD-1, fucose, or CD25. Detection of PD-1 at day 2 by camrelizumab depended on the fucose content (*P* = 0.02), whereas neither pembrolizumab (*P* = 0.7) nor the anti-CD25 antibodies (*P* = 0.18) were affected by changes in fucosylation. Staining with LCA lectin was used to monitor the fucose content at the cell surface (day 2, *t* test, *P* = 0.03). Vehicle refers to DMSO used for 2FPF solubilization. Average values of the geometric mean fluorescence intensity and SD of cells from three donors are represented.

**Table 1. Affinity analysis of anti-PD-1 antibodies for PD-1 variants.**

| | $K_D$ (M) | | | |
|---|---|---|---|---|
| | **Camrelizumab** | **Cemiplimab** | **Pembrolizumab** | **Nivolumab** |
| PD-1 | $3.41 \times 10^{-9}$ | $2.95 \times 10^{-9}$ | $2.25 \times 10^{-9}$ | $4.96 \times 10^{-9}$ |
| PD-1 NF | $3.80 \times 10^{-7}$ | $8.71 \times 10^{-8}$ | $3.67 \times 10^{-9}$ | $2.08 \times 10^{-8}$ |
| PD-1 N58Q | $4.49 \times 10^{-7}$ | $2.60 \times 10^{-7}$ | $2.96 \times 10^{-9}$ | $1.07 \times 10^{-8}$ |

Camrelizumab and cemiplimab exhibited substantially a weaker binding affinity to PD-1 NF compared with PD-1 ($P = 0.02$ for camrelizumab and $P = 0.0003$ for cemiplimab); pembrolizumab showed no difference ($P = 0.08$); and nivolumab exhibited a marginal difference ($P = 0.01$).

binding of camrelizumab in a stronger way compared with non-fucosylated PD-1 (Fig 4C). Similarly, the PD-1:PD-L1 blocking activity of camrelizumab was affected by fucosylated PD-1 to a larger extent than non-fucosylated PD-1 (Fig 4D). Altogether, these data indicate that there is variability in serum PD-1 fucosylation of cancer patients and that this heterogeneity might contribute to the efficacy of immune checkpoint inhibitors in patients.

## Discussion

Glycosylation impacts the structure of proteins, their activity and interactions with other molecules (28, 29). The availability of increasingly sophisticated tools to characterize this protein modification has allowed the definition of structure–function relationships, bringing light to the complexity of the glycan structures, the fine regulation of the underlying biosynthetic mechanisms, and their detailed roles (30, 31, 32, 33, 34). For example, modulation of IgG1 binding to FcγRIIIA by Fc fucosylation is a well-known feature that has been exploited in therapeutic design to enhance antibody-dependent cytotoxicity and antibody-dependent phagocytosis (35). Although currently underexplored, glycoprofiling of new or validated drug targets represents a valuable opportunity for the development of next-generation therapeutics that may achieve exquisite selectivity by binding to distinct glycosylation variants. In addition, the recent discovery of fucosylated biomarkers in the blood of melanoma patients that failed to achieve extended survival after treatment with immune checkpoint inhibitors points to a role of glycosylation not only in the tumor but also in the periphery (26, 27). Therefore, glycosylation analysis of circulating glycoproteins may offer crucial information for both drug design and biomarker discovery.

In this work, first, we observed that camrelizumab interacted with the core fucose of the N58 N-glycan of PD-1. We then confirmed that camrelizumab and cemiplimab bound preferentially to PD-1 carrying glycans with the core fucose. Camrelizumab exhibited a stronger dependence on this modification compared with cemiplimab, probably because of the contribution of serine S31 in the HCDR1 in binding to fucose. Using a combination of protein- and cell-based assays, we showed that the ligand blocking activity of these two antibodies is sensitive to variations in fucose. Lastly, we provided preliminary data indicating that fucosylation of soluble PD-1 in blood varies in different stages of lung cancer, and it is elevated in stage IV. As soluble PD-1 competes with cell-bound PD-1 for binding to antibodies, fucosylation may drive distinct binding

patterns and activity of anti-PD-1 antibodies. Other antibodies that depend on the N58 glycan for binding, including MW11-h317, mAB059c, and STM418, might be regulated in a similar fashion by the glycan structure (36, 37, 38).

Fucosylation can regulate PD-1 function in T cells. Whereas both fucosylated and non-fucosylated forms of PD-1 can be found on T cells, the stability of PD-1 may be enhanced by fucosylation at the sites N49 and N74 (13). It is interesting that core fucosylation promotes the stability of the cell surface adhesin LCAM1, likely by blocking shedding caused by proteases (39). Similarly, MHC-II fucosylation accumulates this protein at the surface of tumor cells (40). Therefore, core fucosylation might exhibit a general role in supporting receptor stability at the cell surface. Interestingly, as fucosylation leads to PD-1 accumulation on the T cell surface and limits activation of murine T cells, it has been suggested that exhausted cells display high levels of fucosylated PD-1. As there is debate around which subsets of exhausted T cells (precursor or terminally exhausted) can be reactivated by anti-PD-1 antibodies, glycoform-specific antibodies may help provide insights on this matter.

First, this work shows that fucosylation of PD-1 has an impact on binding of clinically used blocking antibodies. In addition to the fucosylation at site N58, it is possible that other glycans may have an impact on antibody binding. Moreover, whereas we observed a 100-fold difference in dissociation constant values between fucosylated and non-fucosylated PD-1 for camrelizumab, it resulted in a fivefold to 10-fold difference in PD-1:PD-L1 blocking activity. Then, additional studies in primary cells with endogenous PD-1 expression will be needed to provide a full understanding of the functional relevance of fucosylation. Furthermore, the lack of data from clinical trials comparing camrelizumab or cemiplimab with glycosylation-independent anti-PD-1 antibodies does not allow a clear assessment of the additional benefit of targeting fucosylated PD-1. Preclinical and clinical evidence suggests that a significant overall survival rate across multiple tumor indications is obtained with camrelizumab (41, 42), even if pharmacodynamics data from human phase 1 trials of camrelizumab and nivolumab revealed comparable PD-1 binding affinity and receptor occupancy on circulating T lymphocytes (5, 43). Of note, whereas camrelizumab displayed similar rates of grade 3–4 treatment-related adverse events compared with nivolumab and pembrolizumab, ranging from 22 to 25% of the cohorts studied, reactive cutaneous capillary endothelial proliferation was observed much more frequently with camrelizumab compared with nivolumab and pembrolizumab (67.0% versus 2.4%) (44). This has been attributed to the unexpected binding to vascular endothelial growth factor receptor 2, frizzled

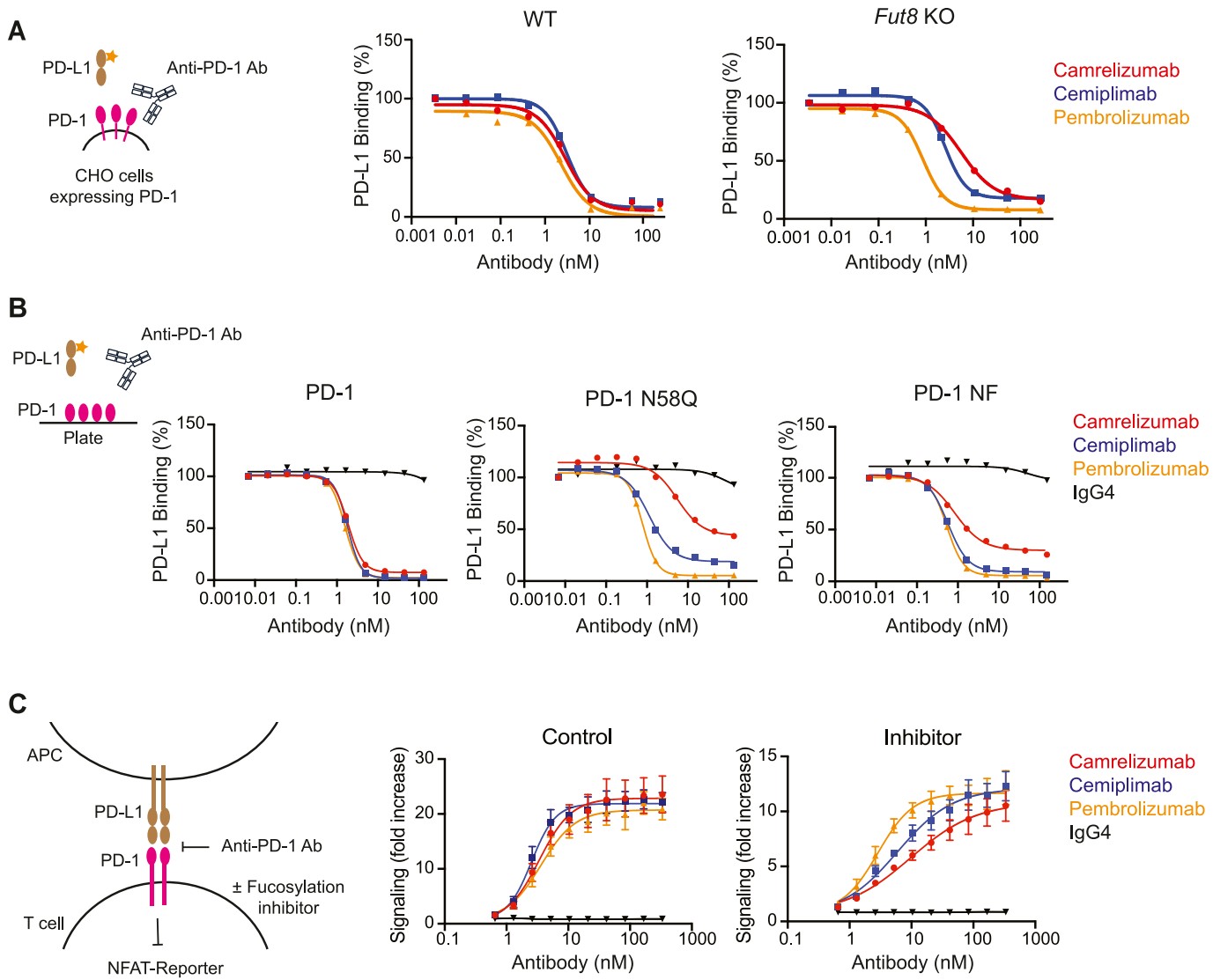

**Figure 3. PD-1 fucosylation impacts the ability of camrelizumab and cemiplimab to block PD-1:PD-L1 interactions.**
**(A)** Blockade of PD-L1 binding to PD-1 expressed in CHO-K1 (WT) or CHO-K1 *Fut8* KO cells by camrelizumab, cemiplimab, or pembrolizumab. PD-L1 binding was assessed by flow cytometry. Three independent experiments were conducted, with each assay performed in duplicate as technical replicates. Average values are represented. Interpolation was calculated using a four-parameter logistic function. **(B)** Blockade of PD-L1 binding to recombinant by anti-PD-1 antibodies was assessed by a plate-based assay. Average values of two independent experiments, each performed in duplicate, are represented. Interpolation was calculated using a four-parameter logistic function. **(C)** Reduction in fucosylation alters the ability of PD-1 antibodies to modulate PD-1 signaling. A schematic diagram of the PD-1/PD-L1 cell-based assay, which relies on a reporter T-cell line engineered to express PD-1, TCR, and CD28, and an APC line expressing PD-L1, antigen–MHC, and CD80. Camrelizumab and cemiplimab blocking activity is affected in the absence of fucose ($P = 0.016$ for camrelizumab and $P = 0.005$ for cemiplimab). Pembrolizumab blocking activity was not modified by the fucosylation status of PD-1 ($P = 0.4$). Luciferase activity is used as an indicator of PD-1 signaling activity, which is blocked by interaction with PD-L1 expressed by the APC. Luminescence signals are represented as a fold increase over non-induced cells. Average values and the standard error of the mean of three independent assays, each performed in duplicate as technical replicates, are represented. Interpolation was calculated using a four-parameter logistic function.

class receptor 5, and UL16 binding protein 2 that may correlate with the side effects of capillary hemangiomas observed in clinical studies with camrelizumab (45). Lastly, the preliminary observation of an elevated concentration of fucosylated PD-1 in serum warrants studies in large patient cohorts.

In summary, we demonstrate that two anti-PD-1 antibodies currently used in a clinical routine prefer glycosylated PD-1 modified with fucose, a frequent but variable element in glycoproteins. Future studies should determine the PD-1 glycosylation

status both in T cells in different activation states and in the blood of patients scheduled to receive immune checkpoint inhibitor therapy. Methods with increased sensitivity, such as glycoproteomics, may prove particularly useful for such analyses. This research supports a framework for the development of antibody therapeutics that target distinct subsets of target proteins. If the glycosylation status of membrane-bound and soluble targets is comparable, measuring glycosylation of the circulating form provides a convenient strategy to obtain information about the tumor

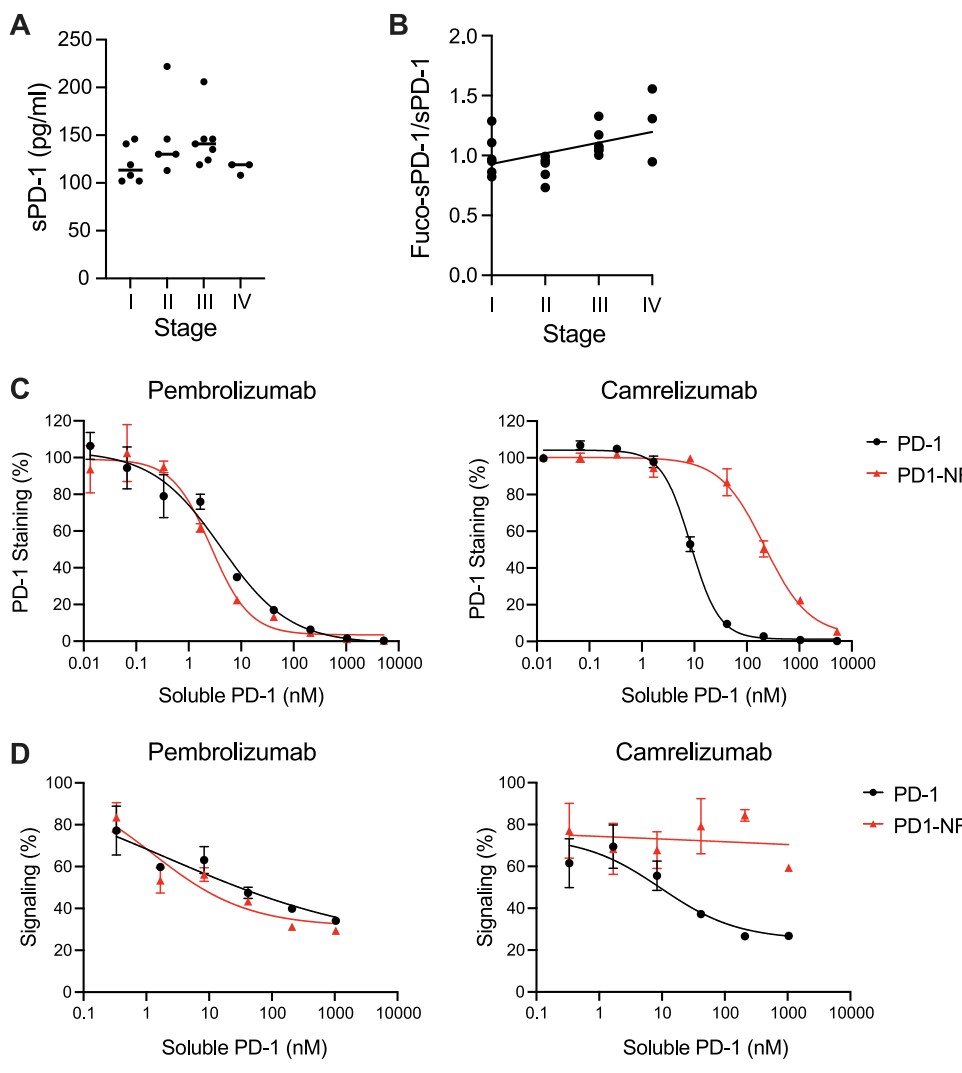

**Figure 4. Fucosylation of serum PD-1 of NSCLC patients varies in different disease stages and modifies the blocking activity of camrelizumab.**
**(A)** Concentration of sPD-1 does not significantly change during progression of disease in this cohort of NSCLC patients. Bars represent average values. **(B)** Fucosylated sPD-1 is higher in the serum of late-stage lung cancer patients. The fucosylated sPD-1 fraction was calculated as the ratio of fucosylated sPD-1 to total sPD-1 concentration. Data were interpolated with linear regression with a slope of 0.089. The deviation from zero was significant with a $P$-value of 0.025. Fucosylation of sPD-1 was found to be elevated in stage IV samples compared with the other stages ($P < 0.0001$, one-sample $t$ test). **(C)** Soluble PD-1 competes with cell-bound PD-1 for binding to anti-PD-1 antibodies. The fucosylation content of soluble PD-1 does not alter the blocking pattern of pembrolizumab ($P = 0.19$) but modifies the binding pattern of camrelizumab ($P = 0.014$). Binding of anti-PD-1 antibodies to Jurkat cells expressing PD-1 was determined by flow cytometry (n = 3). **(D)** Fucosylation status of soluble PD-1 affects the blocking activity of camrelizumab. The effects of soluble PD-1 variants were monitored using a PD-1/PD-L1 cell-based assay in the presence of 2 μg/ml camrelizumab or pembrolizumab.

microenvironment. Conversely, if peripheral glycosylation and tissue glycosylation differ, this feature may be engineered in the antibody specificity to enhance tumor targeting or avoid peripheral sink effects.

# Materials and Methods

### Reagents

2FPF was purchased from Cayman Chemicals and dissolved in DMSO (Cell Signaling Technologies). The anti-human PD-1 antibodies camrelizumab, pembrolizumab, nivolumab, and cemiplimab were from Selleck Chemicals. The human IgG4 isotype was from BioLegend.

### Serum samples

Pretreatment NSCLC serum samples were sourced from iSpecimen. Samples were collected in accordance with relevant applicable guidelines and regulations for human subjects' protection as outlined in the Declaration of Helsinki. All protocols were approved under respective Institutional Review Boards and Ethics Committees (20223899). All subjects used in this study provided written informed consent before collection of samples. The collected clinical data included age at blood draw, body mass index, sex, race, and histopathological data and clinical staging, when applicable.

### Cell lines and culture conditions

CHO-K1 and CHO-K1 *Fut8* KO cell lines (Creative Biogene) were cultured in RPMI 1640 media (Thermo Fisher Scientific) supplemented with 10% FBS and 1% penicillin–streptomycin. Jurkat-Lucia TCR-hPD-1 and Raji-APC-hPD-L1 cells (InvivoGen) were cultured in IMDM (Thermo Fisher Scientific) supplemented with 10% FBS (VWR) and appropriate antibiotics (InvivoGen). Cells were maintained in a humidified incubator at 37°C with 5% $CO_2$. Expi293 cells (Thermo Fisher Scientific) were grown in Expi293 expression media (Thermo Fisher Scientific) in a humidified incubator at 37°C with 8% $CO_2$ at 125 rpm (25 mm shaking throw).

## Recombinant protein production

DNA encoding for residues 24–167 of the extracellular portion of human PD-1 (UniProt Q15116) with a C-terminus histidine tag or a corresponding PD-1 N58Q mutant was cloned into pcDNA3.4 by GenScript. Expi293 cells were transiently transfected with plasmid DNA mixed with PEI (Polysciences) and incubated for 5 d. For the expression of non-fucosylated PD-1, Expi293 cells were grown in the presence of 0.6 mM 2FPF. PD-1 proteins were purified by affinity chromatography using Ni Sepharose Excel resin (Cytiva). PD-10 columns (Cytiva) were used to exchange the buffer to PBS (Corning). Protein quality and purity were assessed by SDS–PAGE (Bio-Rad) and size-exclusion chromatography using a Superdex 200 5/150 column (Cytiva) with ACQUITY UPLC (Waters). The protein concentration was determined using a NanoDrop spectrophotometer.

## Mass spectrometry analysis of recombinant PD-1 variants

For chymotrypsin digestion, 5 µg of each PD-1 variant was diluted with 50 mM ammonium bicarbonate buffer (Sigma-Aldrich) and reduced with the addition of 5 mM DTT (Sigma-Aldrich). Samples were incubated at 60°C for 20 min and then alkylated with 10 mM iodoacetamide (Sigma-Aldrich) in the dark at 25°C for 1 h. Sequencing-grade chymotrypsin (Promega) was added at a protein: enzyme ratio of 40:1, and the samples were incubated at 25°C for 8 h. For trypsin digestion, 5 µg of each PD-1 variant was diluted in a 50 mM ammonium bicarbonate buffer (Sigma-Aldrich) containing 5 mM DTT. Samples were incubated at 60°C for 45 min and then alkylated with 10 mM iodoacetamide for 30 min at 25°C. Mass spectrometry–grade trypsin (Pierce) was added to the samples at a protein:enzyme ratio of 40:1. The samples were then incubated at 37°C for 8 h before LC-MS–grade formic acid (LiChropur) was added to a final concentration of 1% to quench the reaction.

Digested samples were analyzed using an Orbitrap Exploris 480 mass spectrometer (Thermo Fisher Scientific) after desalting with an Acclaim PepMap 100 C18 trap column (Thermo Fisher Scientific), and liquid chromatography separation with an UltiMate 3000 RSLCnano system (Dionex). Samples eluting from the LC were ionized with Nanospray Flex Ion Source at a spray voltage of 2800 V. Each sample was acquired using a top 20 data-dependent acquisition method. Raw data were analyzed with the Glyco-Decipher software suite (46) with trypsin or chymotrypsin set as a protease, depending on the experiment, allowing up to three missed cleavages. Cysteine carbamidomethylation was set as a fixed modification, and methionine oxidation as a variable. Spectrum expansion was enabled, and the other settings were left as default. GlyTouCan (47) was used as a glycan database. Identified spectra were manually validated for correct sequence and glycan assignment, and quantitation was performed based on summed peak areas of the elution profiles of each glycopeptide at all the detected charge states.

## Antibody binding assays

The binding kinetics of PD-1 antibodies were determined using an Octet RED96 instrument (ForteBio), following the previously described procedures (48). All proteins were diluted in Octet kinetics buffer (Sartorius). Antibodies were captured on anti-human Fc AHC2 tips (Sartorius). Data capture was performed using Octet Data Acquisition software, version 9.0 (ForteBio). Data analysis was performed using Octet Data Analysis software, version 9.0 (ForteBio), using a 1:1 binding model.

Anti-PD-1 antibody binding was also assayed by flow cytometry using Jurkat-Lucia TCR-hPD-1 cells or CHO-K1 and CHO-K1 *Fut8* KO cells transfected with plasmids encoding for human PD-1 or chimeric receptors including the extracellular and transmembrane domain of PD-1 linked to the intracellular green fluorescent protein (GenScript) by ExpiFectamine (Thermo Fisher Scientific). Anti-PD-1 antibodies or human IgG4 isotypes were added to the cells and incubated for 30 min on ice. Cells were washed and stained with 1 µg/ml PE-conjugated anti-human IgG4 (SouthernBiotech) or 3 µg/ml Alexa Fluor 647–conjugated anti-human IgG (Jackson ImmunoResearch) antibodies for 30 min. Cells were analyzed using a CytoFLEX instrument (Beckman Coulter).

In addition, anti-PD-1 antibody binding was assessed by flow cytometry using human naive CD8 T cells. Two million CD8 T cells (STEMCELL Technologies) from three donors were cultured for 48 h in AIM V Medium (Thermo Fisher Scientific) supplemented with 5% heat-inactivated FBS and activated with Dynabeads Human T-Activator CD3/CD28 (Thermo Fisher Scientific) at a 1:1 bead:cell ratio according to the manufacturer's instructions, in the presence or absence of 0.1 mM of 2FPF. At days 1 and 2 after stimulation, cells were stained with a Zombie Red viability dye kit (BioLegend), followed by incubation in fixation solution (eBioscience–Thermo Fisher Scientific). Core fucosylation was assessed by incubating cells with FITC-conjugated LCA lectin (Vector Labs) at a 1:500 dilution. To confirm activation followed by bead stimulation, cells were stained with BV711-conjugated anti-CD25 antibody (Clone M-A251; BioLegend) at a 1:50 dilution in PBS. Cells were incubated with pembrolizumab, camrelizumab, or human IgG4 isotype control (at 1 nM concentration), followed by incubation with PE-conjugated anti-human IgG4 at a 1:300 dilution in PBS. Cells were analyzed by a CytoFLEX flow cytometer.

## Ligand blocking assays

To assess the binding of PD-L1 to PD-1, biotinylated human PD-L1 (ACROBiosystems) and PE-conjugated streptavidin (BioLegend) were mixed at a 2:1 M ratio, added to PD-1–transfected CHO-K1 or CHO-K1 *Fut8* KO cells, and incubated for 20 min on ice. Cells were washed and analyzed by a CytoFLEX flow cytometer. To investigate the blocking of PD-L1 binding, CHO-K1 and CHO-K1 *Fut8* KO cells transfected with plasmids for PD-1 expression were incubated with antibodies for 15 min on ice before the addition of PD-L1:SA-PE complexes. Cells were then washed and analyzed by a CytoFLEX flow cytometer.

For plate-based ligand blocking assay, 96-well flat-bottom plates (Thermo Fisher Scientific) were coated with 5 µg/ml of purified recombinant PD-1, PD-1 NF, or PD-1 N58Q in PBS overnight at room temperature. Plates were then washed with PBS containing 0.05% Tween-20 (PBST). After blocking with 5% BSA (MilliporeSigma) in PBS for 1 h at room temperature, wells were washed with PBST, followed by incubation with anti-PD-1 antibodies diluted in PBS containing

0.5% BSA for 1 h at room temperature. After washing with PBST, 1 µg/ml PD-L1-Fc-biotin in PBS containing 0.5% BSA was added to each well and incubated for 2 h. After washing with PBST, the wells were incubated with 0.1 µg/ml horseradish peroxidase-conjugated streptavidin (ACROBiosystems) for 30 min. Wells were washed and incubated with a 3,3′,5,5′-tetramethylbenzidine substrate (BioLegend), followed by a stop solution (BioLegend). Absorbance at 450 nm was read with an EnVision microplate reader (PerkinElmer).

### PD-1:PD-L1 functional assay

Jurkat TCR-hPD-1 cells were precultured in a medium with or without 300 µM 2FPF for 3 d. Antibodies and PD-1 proteins were diluted in PBS and added to 96-well flat-bottom plates. Jurkat-Lucia TCR-hPD-1 cells and Raji-APC-hPD-L1 were added to the wells according to the manufacturer's directions, and plates were incubated at 37°C for 6 h. Supernatants were transferred into a 96-well white plate (VWR) and assayed with QUANTI-Luc 4 reagent (InvivoGen). Luminescence was read with an Envision system.

### Serum PD-1 quantification

Serum and recombinant PD-1 were quantified by a human PD-1 Quantikine ELISA and a human PD-1 DuoSet ELISA (R&D Systems), following the manufacturer's protocols. In addition, serum and recombinant PD-1 were quantified by a sandwich ELISA using wells coated overnight with 5 µg/ml camrelizumab or pembrolizumab in PBS, followed by incubation with 1% BSA in PBS. After sample incubation, PD-1 was detected using the human PD-1 DuoSet ELISA detection antibody. Wells were washed and incubated with a 3,3′,5,5′-tetramethylbenzidine substrate. The reaction was stopped with acid solution, and absorbance was read with an Envision microplate reader.

### Data analysis and visualization

Analysis of flow cytometry data was performed using FlowJo 10.9.0 (FlowJo). GraphPad Prism 10 (GraphPad Software) was used to represent data and for statistical analysis.

### Protein structure analysis

Interactions and interfaces between antibodies or ligands and PD-1 were calculated with PDBePISA (49) and CoCoMaps (50) using PDB structures 7CU5 (16), 7WVM (17), 4ZQK (51), 5WT9 (21), and 5B8C (52). Structures were visualized by PyMOL (Schroedinger, Inc.).

## Supplementary Information

## Acknowledgements

We thank all current and past colleagues at InterVenn for input on the project, and Timothy Ohara for article review. We are grateful to Carlito Lebrilla and Carolyn Bertozzi for scientific discussions and feedback.

## Author Contributions

C-W Chu: data curation, formal analysis, investigation, methodology, and writing—original draft, review, and editing.
T Čaval: data curation, formal analysis, investigation, methodology, and writing—review and editing.
F Alisson-Silva: data curation, formal analysis, investigation, methodology, and writing—review and editing.
A Tankasala: data curation, formal analysis, methodology, and writing—review and editing.
C Guerrier: data curation and writing—review and editing.
G Czerwieniec: data curation, formal analysis, investigation, methodology, and writing—review and editing.
H Läubli: conceptualization and writing—original draft, review, and editing.
F Schwarz: conceptualization, data curation, formal analysis, supervision, investigation, methodology, and writing—original draft, review, and editing.

## Conflict of Interest Statement

C-W Chu, T Čaval, F Alisson-Silva, A Tankasala, C Guerrier, G Czerwieniec, and F Schwarz are or were employees of InterVenn Biosciences, a company that applies glycoproteomics and artificial intelligence to discover biomarkers and develop diagnostic tests. H Läubli is a consultant of InterVenn Biosciences and has received travel grants and consultant fees from Bristol-Myers Squibb, Alector, and MSD. H Läubli has received research support from Bristol-Myers Squibb, Novartis, GlycoEra, and Palleon Pharmaceuticals.

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
