## [Reviewer comments · Life Science Alliance]

Life Science Alliance

Variable PD-1 glycosylation modulates the activity of immune checkpoint inhibitors

Chih-Wei Chu, Tomislav Čaval, Frederico Alisson-Silva, Akshaya Tankasala, Christina Guerrier, Gregg Czerwieniec, Heinz Läubli, and Flavio Schwarz

DOI: <https://doi.org/10.26508/lsa.202302368>

Corresponding author(s): Flavio Schwarz, InterVenn Biosciences

Review Timeline:

Submission Date:	2023-09-12
Editorial Decision:	2023-10-27
Revision Received:	2023-12-11
Editorial Decision:	2023-12-13
Revision Received:	2023-12-19
Accepted:	2023-12-20

Transaction Report:

October 27, 2023

Re: Life Science Alliance manuscript #LSA-2023-02368-T

Flavio Schwarz
InterVenn Biosciences
2 Tower Pl 5th floor
CA 94080

Dear Dr. Schwarz,

Thank you for submitting your manuscript entitled "Variable PD-1 glycosylation modulates the activity of immune checkpoint inhibitors" to Life Science Alliance. The manuscript was assessed by expert reviewers, whose comments are appended to this letter. We invite you to submit a revised manuscript addressing the Reviewer comments.

Thank you for this interesting contribution to Life Science Alliance. We are looking forward to receiving your revised manuscript.

Sincerely,

B. MANUSCRIPT ORGANIZATION AND FORMATTING:

Reviewer #1 (Comments to the Authors (Required)):

"Fucosylation on surface receptors plays a role in facilitating cancer cell growth and also contributes to the tumor microenvironment. In this study, the authors show the impact of fucosylation on PD-1, a critical surface receptor associated with immune response regulation. Specifically, the authors examine how the absence of fucosylation PD1, achieved through FUT8 KO cells and the Fucosyltransferase Inhibitor 2FPF, influences the interaction between fucosylated PD-1 and PD-1 antibodies. Fucosylated PD-1 impact the ability of camrelizumab and cemiplimab to block PD-1/PD-L1 interactions. The study also highlights the elevated fucosylation of soluble PD-1 (sPD-1) in the serum of individuals with non-small cell lung cancer (NSCLC) compared to different stages of the disease. This study carries significant importance as it introduces the concept of antibodies targeting specific glycosylation sites to enhance selectivity.

This work is important and could help improve the field of immunotherapy. However, a notable concern arises from the similarity of these findings to a previous study (Blockage of Core Fucosylation Reduces Cell-Surface Expression of PD-1 and Promotes Anti-tumor Immune Responses of T Cells, Cell Rep . 2017 Aug 1;20(5):1017-1028)

To address this concern, it would be beneficial for the authors to incorporate more cell-based data demonstrating that blocking core fucosylation enhances anti-tumor immunity instead of antibody binding. It may carry out cell-based assays to assess T-cell exhaustion, cytokine levels, etc. While the current study shows that removing fucose from PD-1 reduces its binding with PD-1 antibodies in Octet assays and FUT8 KO cells, Fut8 KO affects core fucosylation across all proteins. Additionally, the authors emphasized the fucosylated sPD1 in NSCLC was higher. It can pave the way towards the implementation of personalized therapeutic approaches. Does fucosylated sPD1 affect PD1 antibodies binding to membrane PD1? It may explore or describe more the differential response between sPD-1 and cell-bound PD-1 regarding PD-1 antibodies binding in patient samples.

In summary, the findings presented in this study are important, addressing the concerns related to previous research and providing more cell-based data could enhance its impact.

minor revisions:

1. PD-1 NF is non-fucosylated PD-1 or PD-1 non- fucosylated, what this abbreviation for? it may add the full name of PD-1 NF in the end of the figure legend, ex, Fig 2,3...
2. line 45:"only a fraction of patients benefit from these drugs and develop durable clinical responses..." It may add one or two sentences to describe why some patients would not respond to PD-1/PD-L1 interaction during PD1 antibody treatment. ex: exhausted T cells
3. in line 157: (Cells were washed and stained with a 0.1 mg/l anti-human IgG Fc-PE antibody for 20 157 minutes.) 0.1mg/l was used in flow binding assay, please make sure it is the right condition.
4. fig.3C and D: show binding profiles of anti-PD-1 antibodies to recombinant PD-1 proteins or in PD1 stable cell lines, in line 430, in your statement" Lack of fucosyltransferase FUT8 expression that adds core fucose does not alter the binding profile of pembrolizumab but results in reduced binding by cemiplimab and camrelizumab..... " It shows that loss of fucose in PD1 affects cemiplimab and camrelizumab. However, from the binding curve in Fig 3D, it appears that pembrolizumab increases binding from PD1 to PD-1 NF (from fut8 KO cells).
5. In Supplementary Fig 7, how long did the inhibitor act on the cells to determine cell survival, please describe the incubate time in fig legend.

Reviewer #2 (Comments to the Authors (Required)):

The current manuscript describes fucosylated PD-1-selective binding by some of the clinically approved anti-PD-1 antibodies. These antibodies were previously reported to recognize glycosylated PD-1. The current study identified fucose within the sugar chain as a crucial component of antibody binding. This is a solid result as evidenced from multiple experimental systems using Fut8-knockout, a fucosylation inhibitor and a PD-1 mutant. The significance of PD-1 glycosylation in T cell subsets is yet to be explored due to the methodological limitations. Antibodies that can discriminate different glycosylation status may be a precious tool in future studies.

The authors developed the ELISA-based detection system of fucosylated/non-fucosylated human PD-1. Using this system, they

examined both PD-1 forms in serum samples from cancer patients (Fig. 4). The increase of fucosylated PD-1 in late-stage patients is claimed, but the increasing trend is not particularly clear. The method and result of stats calculation should be provided. Stats information should be provided in all Figures and Tables.

A number of items in the reference list lack some important information such as journal name, volume and page numbers.

Reviewer #3 (Comments to the Authors (Required)):

The authors (and in particular Flavio Schwarz) are experts in the field of protein glycosylation and in the pharmaceutical sector. PD-1 blocking antibodies represent a leading class of anti-cancer therapeutics.

The discovery made in this paper, clearly documented both by BIAcore data and by ingenious glycosylation engineering of PD-1, is that this immune checkpoint protein is glycosylated with variable glycostructures and that the binding of some monoclonal antibodies (e.g., cemiplimab), routinely used in the clinical practice, is greatly influenced by the presence or absence of a core fucose in Asparagine 58 of PD-1.

These findings are clinically relevant, as corroborated by the analysis of glycovariants in the antigen derived from patients with SCLC.

The authors are to be commended for this manuscript, which combined a rigorous experimental work with a clear and concise presentation.

Reviewer #1 (Comments to the Authors (Required)):

"Fucosylation on surface receptors plays a role in facilitating cancer cell growth and also contributes to the tumor microenvironment. In this study, the authors show the impact of fucosylation on PD-1, a critical surface receptor associated with immune response regulation. Specifically, the authors examine how the absence of fucosylation PD1, achieved through FUT8 KO cells and the Fucosyltransferase Inhibitor 2FPF, influences the interaction between fucosylated PD-1 and PD-1 antibodies. Fucosylated PD-1 impact the ability of camrelizumab and cemiplimab to block PD-1/PD-L1 interactions. The study also highlights the elevated fucosylation of soluble PD-1 (sPD-1) in the serum of individuals with non-small cell lung cancer (NSCLC) compared to different stages of the disease. This study carries significant importance as it introduces the concept of antibodies targeting specific glycosylation sites to enhance selectivity.

This work is important and could help improve the field of immunotherapy. However, a notable concern arises from the similarity of these findings to a previous study (Blockage of Core Fucosylation Reduces Cell-Surface Expression of PD-1 and Promotes Anti-tumor Immune Responses of T Cells, Cell Rep . 2017 Aug 1;20(5):1017-1028)

To address this concern, it would be beneficial for the authors to incorporate more cell-based data demonstrating that blocking core fucosylation enhances anti-tumor immunity instead of antibody binding. It may carry out cell-based assays to assess T-cell exhaustion, cytokine levels, etc.

While the current study shows that removing fucose from PD-1 reduces its binding with PD-1 antibodies in Octet assays and FUT8 KO cells, Fut8 KO affects core fucosylation across all proteins. Additionally, the authors emphasized the fucosylated sPD1 in NSCLC was higher. It can pave the way towards the implementation of personalized therapeutic approaches. Does fucosylated sPD1 affect PD1 antibodies binding to membrane PD1? It may explore or describe more the differential response between sPD-1 and cell-bound PD-1 regarding PD-1 antibodies binding in patient samples.

In summary, the findings presented in this study are important, addressing the concerns related to previous research and providing more cell-based data could enhance its impact.

We thank the Reviewer for evaluating our manuscript and for highlighting the relevance of this work towards the improvement of the immunotherapy field. We appreciate their suggestions and would like to address the points raised.

Point 1

While we agree that the reference cited by the reviewer is important, we strongly believe that our work describes a significantly different concept. In fact, the referenced study demonstrates that blockade of core fucosylation results in reduced PD-1 expression at the cell surface. Instead, our study shows that PD-1 fucosylation status determines the binding of different anti-PD-1 antibodies, influencing their efficacy in blocking PD-1/PD-L1 interaction.

The idea that core fucosylation may modify PD-1 expression is certainly interesting. While we don't intend to discuss this in the manuscript as it would compromise a clear and concise presentation of the data (which was appreciated by Reviewer 3), it is important to mention that we haven't noticed a significant dependence of fucosylation on PD-1 expression in our assays. In fact, activated CD8 T cells appear to have a comparable amount of PD-1 when stimulated in presence or absence of the fucose inhibitor (figure 2D). Similarly, expression of PD-1 in Jurkat cells was not affected by the fucosylation inhibitor (figure S8). There might be many reasons for these differences (including, perhaps, species-specific difference: we used only human cells in our study, whereas the study cited by the Reviewer were carried out in mouse cells). This is, however, outside of the scope of the current work.

While we acknowledge that additional cell-based or *in vivo* assays might further show that PD-1 fucosylation modulates the efficacy of these antibodies in enhancing T cell-mediated anti-tumor immunity, we believe that these data would not change at all the overall conclusions of this study. Thus, also based on the comments of Reviewers 2 and 3, we feel that the current set of biophysical, cell-based and functional data conclusively demonstrate the role of PD-1 fucosylation in binding of two antibodies in a satisfactory way.

Point 2

We thank the Reviewer for bringing up this interesting point. To address the question, we set out two experiments to demonstrate how soluble versions of PD-1 may affect antibody binding to PD-1 at the cell surface.

In the first experiment, we used fucosylation variants of recombinant PD-1 as a surrogate of serum sPD-1 to test how they modify binding of the anti-PD-1 antibodies and showed that binding of camrelizumab is affected in a much more prominent way by fucosylated PD-1 compared to non fucosylated PD-1. In line with the other data, the glycosylation structure of soluble PD-1 did not affect pembrolizumab binding. We then tested the ability of fucosylation variants of soluble PD-1 to alter the efficacy of PD-1:PD-L1 blockade in a functional assay and found a comparable effect. We added these data in Figure 4B and 4C, along with appropriate descriptions.

minor revisions:

1. PD-1 NF is non-fucosylated PD-1 or PD-1 non-fucosylated, what this abbreviation for? it may add the full name of PD-1 NF in the end of the figure legend, ex, Fig 2,3...

"PD-1 NF" is recombinant, non-fucosylated PD-1. We added this label in the figure legends, as appropriate.

2. line 45:"only a fraction of patients benefit from these drugs and develop durable clinical responses..." It may add one or two sentences to describe why some patients would not respond to PD-1/PD-L1 interaction during PD1 antibody treatment. ex: exhausted T cells

We added a sentence describing the mechanism that may lead to limited efficacy of immunotherapy. “Many mechanisms may contribute to limit this approach: the lack of appropriate T cell priming, their exhaustion status, the presence of stromal factors that limit penetration of T cells to the tumor parenchyma, and the relative abundance of lymphoid and myeloid cells with immunosuppressive properties”.

3. in line 157: (Cells were washed and stained with a 0.1 mg/l anti-human IgG Fc-PE antibody for 20 157 minutes.) 0.1mg/l was used in flow binding assay, please make sure it is the right condition.

We corrected this in the text. The concentration used was 1 mg/l.

4. fig.3C and D: show binding profiles of anti-PD-1 antibodies to recombinant PD-1 proteins or in PD1 stable cell lines, in line 430,
in your statement" Lack of fucosyltransferase FUT8 expression that adds core fucose does not alter the binding profile of pembrolizumab but results in reduced binding by cemiplimab and camrelizumab..... " It shows that loss of fucose in PD1 affects cemiplimab and camrelizumab. However, from the binding curve in Fig 3D, it appears that pembrolizumab increases binding from PD1 to PD-1 NF (from fut8 KO cells).

The Reviewer probably refers to Figure 2 here, as the statement “Lack of fucosyltransferase FUT8 expression that adds core fucose does not alter the binding profile of pembrolizumab but results in reduced binding by cemiplimab and camrelizumab...” refers to Figure 2C. We believe that this was a confounding effect of the transfection efficiency. To control for this, we repeated the experiment using a construct for expression of a chimeric receptor with the extracellular and transmembrane region of PD-1 and intracellular GFP. By gating on GFP-positive cells, we controlled for transfection efficiency and obtained more accurate measurements of PD-1 expression. We updated Figure 2C in the revised manuscript and added an appropriate description.

5. In Supplementary Fig 7, how long did the inhibitor act on the cells to determine cell survival, please describe the incubate time in fig legend.

The cells were treated for 3 days. We added this information in the figure legend.

Reviewer #2 (Comments to the Authors (Required)):

The current manuscript describes fucosylated PD-1-selective binding by some of the clinically approved anti-PD-1 antibodies. These antibodies were previously reported to recognize glycosylated PD-1. The current study identified fucose within the sugar chain as a crucial component of antibody binding. This is a solid result as evidenced from multiple experimental systems using Fut8-knockout, a fucosylation inhibitor and a PD-1 mutant. The significance of PD-1 glycosylation in T cell subsets is yet to be explored due to the methodological limitations. Antibodies that can discriminate different glycosylation status may be a precious tool in future studies.

The authors developed the ELISA-based detection system of fucosylated/non-fucosylated human PD-1. Using this system, they examined both PD-1 forms in serum samples from cancer patients (Fig. 4). The increase of fucosylated PD-1 in late-stage patients is claimed, but the increasing trend is not particularly clear. The method and result of stats calculation should be provided. Stats information should be provided in all Figures and Tables.

We are grateful to the Reviewer for evaluating our manuscript and for the positive feedback.

We added statistics for the data included in figure 4B, as well as for the other figures and tables, as applicable. In addition, to simplify the description of the data, we provided fold changes in EC50 and IC50 values, when possible.

We agree that the current data indicating elevated levels of fucosylated PD-1 are to be considered preliminary, and that studies in larger patient cohorts are needed to validate this initial observation. To account for this, we added a sentence in the discussion to describe this limitation of the current study: “Lastly, the preliminary observation of elevated concentration of fucosylated PD-1 in serum warrants studies in large patient cohorts” (page 14).

A number of items in the reference list lack some important information such as journal name, volume and page numbers.

We added the missing information in the reference list. We included a few additional references, as appropriate.

Reviewer #3 (Comments to the Authors (Required)):

The authors (and in particular Flavio Schwarz) are experts in the field of protein glycosylation and in the pharmaceutical sector. PD-1 blocking antibodies represent a leading class of anti-cancer therapeutics.

The discovery made in this paper, clearly documented both by BIAcore data and by ingenious glycosylation engineering of PD-1, is that this immune checkpoint protein is glycosylated with variable glycostructures and that the binding of some monoclonal antibodies (e.g., cemiplimab), routinely used in the clinical practice, is greatly influenced by the presence or absence of a core fucose in Asparagine 58 of PD-1.

These findings are clinically relevant, as corroborated by the analysis of glycovariants in the antigen derived from patients with SCLC.

The authors are to be commended for this manuscript, which combined a rigorous experimental work with a clear and concise presentation.

We thank the Reviewer for evaluating our manuscript and for the positive feedback. We are confident that this work will both provide valuable information for these antibodies to the scientific community, and also inspire drug developers to utilize receptor glycoprofiling for the development of next generation molecules that targets specific subsets of receptors.

December 13, 2023

RE: Life Science Alliance Manuscript #LSA-2023-02368-TR

Dr. Flavio Schwarz
InterVenn Biosciences
2 Tower Pl 5th floor
South San Francisco, CA 94080

Dear Dr. Schwarz,

Thank you for submitting your revised manuscript entitled "Variable PD-1 glycosylation modulates the activity of immune checkpoint inhibitors". We would be happy to publish your paper in Life Science Alliance pending final revisions necessary to meet our formatting guidelines.

- please consult our manuscript preparation guidelines <https://www.life-science-alliance.org/manuscript-prep> and make sure your manuscript sections are in the correct order
- please add your main, supplementary figure, and table legends to the main manuscript text after the references section
- please add callouts for Figures 1A-C; 2D; S1A-B and S5 your main manuscript text

A. FINAL FILES:

B. MANUSCRIPT ORGANIZATION AND FORMATTING:

****It is Life Science Alliance policy that if requested, original data images must be made available to the editors. Failure to provide**

original images upon request will result in unavoidable delays in publication. Please ensure that you have access to all original data images prior to final submission.**

The license to publish form must be signed before your manuscript can be sent to production. A link to the electronic license to publish form will be available to the corresponding author only. Please take a moment to check your funder requirements.

Sincerely,

December 20, 2023

RE: Life Science Alliance Manuscript #LSA-2023-02368-TRR

Dr. Flavio Schwarz
InterVenn Biosciences
2 Tower Pl 5th floor
South San Francisco, CA 94080

Dear Dr. Schwarz,

Thank you for submitting your Research Article entitled "Variable PD-1 glycosylation modulates the activity of immune checkpoint inhibitors". It is a pleasure to let you know that your manuscript is now accepted for publication in Life Science Alliance. Congratulations on this interesting work.

DISTRIBUTION OF MATERIALS:

Again, congratulations on a very nice paper. I hope you found the review process to be constructive and are pleased with how the manuscript was handled editorially. We look forward to future exciting submissions from your lab.

Sincerely,
